Arbuscular mycorrhizal fungi alter rhizosphere fungal community characteristics of Acorus calamus to improve Cr resistance

Xia Guodong 1
Zhu Sixi zhusixi2011@163.com 1
Zhao Wei 1
Yang Xiuqing 2
Sheng Luying 1
Mao Huan 1
1 Guizhou Minzu University, The Karst Environmental Geological Hazard Prevention of Key Laboratory of State Ethnic Affairs Commission , Guiyang , Guizhou , China
2 Guizhou Minzu University , Guiyang , Guizhou , China
Mora-Montes Héctor
Electronic publication date: 2023 Nov 8
Publication date: 2023
Volume: 11
Electronic Location ID: e15681
Received 2023 Apr 5; Accepted 2023 Jun 13
Copyright: ©2023 Xia et al.
Copyright year: 2023
Copyright holder: Xia et al.
License: This is an open access article distributed under the terms of the Creative Commons Attribution License, which permits unrestricted use, distribution, reproduction and adaptation in any medium and for any purpose provided that it is properly attributed. For attribution, the original author(s), title, publication source (PeerJ) and either DOI or URL of the article must be cited.
License URL: https://creativecommons.org/licenses/by/4.0/

Keywords: Cr stress, Arbuscular mycorrhizal fungi, Fungal community structure, Acorus calamus, ITS rRNA sequencing technology, Phytoremediation

Funding: National Natural Science Foundation of China 31560107 Science and Technology Support Project of Guizhou province, China (Guizhou Branch Support) [2018]2807 This work was supported by the National Natural Science Foundation of China (No. 31560107) and by the Science and Technology Support Project of Guizhou province, China (Guizhou Branch Support [2018]2807). The funders had no role in study design, data collection and analysis, decision to publish, or preparation of the manuscript.

==============================
To investigate changes in fungal community characteristics under different Cr(VI) concentration stresses and the advantages of adding arbuscular mycorrhizal fungi (AMF), we used high throughput sequencing to characterize the fungal communities. Cr(VI) stress reduced rhizosphere soil SOM (soil organic matter) content and AMF addition improved this stress phenomenon. There were significant differences in fungal community changes under different Cr(VI) concentrations. The fungal community characteristics changed through inhibition of fungal metabolic ability, as fungal abundance increased after AMF addition, and the fungal diversity increased under high Cr(VI) concentration. The dominant phyla were members of the Ascomycota, Basidiomycota, Mortierellomycota, and Rozellomycota. Dominant groups relevant to Cr resistance were Ascomycota and Basidiomycota fungi. Moreover, Fungal community characteristics were analyzed using high-throughput sequencing of the cytochrome c metabolic pathway, NADH dehydrogenase, and NADH: ubiquinone reductase and all these functions were enhanced after AMF addition. Therefore, Cr(VI) stress significantly affects fungal community structure, while AMF addition could increase its SOM content, and metabolic capacity, and improve fungal community tolerance to Cr stress. This study contributed to the understanding response of rhizosphere fungal community in AMF-assisted wetland phytoremediation under Cr stress.

Introduction

Nowadays, soil Cr contamination is a global concern (Li et al., 2019a), as Cr is a heavy metal that is harmful to plants and humans (Chen et al., 2018a; Chen et al., 2018b). It is often used in metallurgical industries, electroplating, dyeing agents and some alloy manufacturing (Dhal et al., 2013; Viti & Giovannetti, 2007; Manikandan et al., 2016). Naturally Cr is often found as stable compounds Cr(VI) and Cr(III) (Bartlett, 1991), with Cr(VI) mainly present in soil and water bodies as CrO42− and Cr2O72− (Xia et al., 2019). Cr(VI) is more toxic and more soluble than Cr(III), reported to be more than 100 times more toxic than Cr(III), and can cause DNA damage and carcinogenesis (Garcıa-Hernández, Villarreal-Chiu & Garza-González, 2017; Shi et al., 2018; Wang et al., 2017a; Wang et al., 2017b). Cr enrichment can have a severe impact on human health through the food chain. Therefore, it is crucial to remediate Cr(VI)-contaminated soils

Compared to physicochemical remediation, phytoremediation is a heavy metal remediation technique that has recently been identified as cheap and with no secondary contamination (Xue et al., 2018a; Xue et al., 2018b; Piyush, Singh & Anderson, 2020). Wetland plants, due to their well-developed root systems and vital growth capacity, play an indispensable role in the CWs (constructed wetlands) to treat wastewater containing heavy metals (Jan, Lerika & Jaroslav, 2009). However, studies have shown that high levels of heavy metal pollution can be toxic to wetland plants (Ammara & Saleh, 2020), and so to protect them, arbuscular mycorrhizal fungi (AMF) have been introduced (27 genera in one order, four families and 11 families in the phylum Glomeromycota, with about 300 species) (Redecker et al., 2013). AMF can form symbiotic relationships with the root systems of most terrestrial, aquatic and semi-aquatic plants (Brundrett & Tedersoo, 2018; Calheiros et al., 2019; Nataša & Gaberščik, 2010) which improves soil resources and responds effectively to environmental constraints (Balestrini et al., 2018; Lenoir, Fontaine & Sahraoui, 2016; Torres, Antolín & Goicoechea, 2018). AMF increases plant soil nutrient uptake by colonizing and altering their root systems, while at the same time improving their tolerance to heavy metals by increasing antioxidant enzyme (e.g., catalase, peroxidase, and superoxide dismutase) activity and decreasing reactive oxygen species (ROS) (Lin et al., 2017; Wang et al., 2018; Devi, Gupta & Kapoor, 2019).

There have been numerous studies on bacterial stress to heavy metals (Fan et al., 2017; Sultana et al., 2014), and Cr(VI) reduction by bacteria (Wang et al., 2017a; Wang et al., 2017b; Xia et al., 2018; Viti et al., 2014; Xue et al., 2014), but few studies have addressed mechanisms of Cr(VI) reduction by fungi, especially heterotrophic fungi that are abundant in the soil/plant rhizosphere. Notably, most fungi have heavy metal chelating systems and heavy metal enrichment capabilities (Janoušková, Pavlíková & Vosátka, 2006; Aly, Debbab & Proksch, 2011). Fungal substrates for heavy metal resistance can be divided into intra- and extracellular (Hall, 2002). For example, some fungi can secrete organic acids and amino acids to chelate with heavy metals and thus reduce toxicity (intracellular; Vodnik et al., 2008), while others reduce heavy metal ions through electron transfer (extracellular; Xia et al., 2018). Cr(VI) removal by fungi studies are scarce, but some have shown that fungi also have antioxidant mechanisms to reduce Cr(VI) and thus reduce heavy metal toxic effects on themselves (Viti et al., 2014; Xue et al., 2014; Joutey et al., 2015; Acevedo-Aguilar et al., 2006). Fungal community structure is significantly altered in response to Cr stress (Del Val, Barea & Azcón-Aguilar, 1999; Nordgren, Baath & Soderstrom, 1983), however, few studies have investigated Cr(VI) fungal removal mechanisms and fungal community changes.

In this study, the aim was to compare fungal community structure and functional predictions in an AMF group with a control group using high-throughput sequencing, to reveal fungal response mechanisms after AMF addition under Cr stress, and also provide a basis for further studies on the principle of fungal reduction of Cr(VI). This study will also contribute to our understanding of the response of AMF-assisted wetland phytoremediation to Cr stress in inter-rhizosphere fungal communities. We hypothesized that the fungal community of Acorus calamus root soil would be affected in Cr-stressed soil. However, adding AMF could increase the stress of the fungi community to Cr by increasing SOM content.

Materials and Methods

Material selection

Pots used in the experiments were fully enclosed at the bottom, with bottom length, inner diameter, outer diameter and height of 13.5 cm, 19 cm, 21.5 cm and 14.8 cm, respectively, and were all sterilized. The plants used in this experiment were selected from Acorus calamus, which largely grows in constructed wetlands. Roots were disinfected with 75% alcohol and 1% sodium hypochlorite and then repeatedly rinsed with deionized water. Soil was collected using the five-point sampling method, to a depth of 20 cm, from the barren hills behind the Lost Wetland Research Centre, mixed well and taken back to the laboratory. Large weeds and stones were removed and the soil passed through a two mm sieve, sterilized and used as a culture substrate (pH =6.83). Three kilograms of soil was placed in each sterilized pot. Rhiaophagus intraradices (purchased from Changjiang University) was the AMF used Walker & Schuessler (2010), in a substrate: strain ratio of 5:1.

Experimental design

Experimental groups were divided into AMF-added, and non-AMF-added, each with five Cr concentration levels (0, 10, 50, 100, 200 mg/kg), and three parallel groups for each treatment. Each experimental pot was disinfected with 75% ethanol and 1% sodium hypochlorite solution for 10 s and 15 mins and then carefully cleaned with deionized water five times (Hu et al., 2021). Seedlings were placed in the experimental pots, incubated at a constant temperature of 26 °C and a light intensity of 176 µmol m2 s−1 for a photoperiod of 12 h for 30 days. The greenhouse humidity was stable at 60%rh, then inoculated with AMF and observed after 15 days. After successful planting, the experiment was conducted by adding potassium dichromate in five concentration gradients, followed by Hogaland’s solution (15 mL) and water (150 mL) every 14 d and 3 d, respectively (Zhan et al., 2019).

Sample analysis

At the end of the 30-day experiment, the inter-root soil under the uniformly growing yellow iris was removed, and passed naturally through a two mm sieve to determine soil pH, Electrical Conductivity (EC), SOM, total Cr and Cr(VI). Soil pH was determined by potentiometric titration, soil organic matter (SOM) by oxidation with K2Cr2O7-H2SO4—external heating, and total Cr and Cr(VI) in the soil by extraction with the alkaline solution—flame atomic absorption spectrophotometry method. Another portion of each soil sample was stored in a −20 °C freezer and used for DNA extraction and PCR amplification (Zhao et al., 2023).

DNA extraction and PCR amplification

Total DNA extraction from the microbial community was carried out according to the E.Z.N.A.® soil DNA kit (Omega Bio-Tek, Norcross, GA, U.S.) instructions, and DNA extraction quality was determined using 1% agarose gel electrophoresis. PCR amplification of the ITSrRNA gene was performed using ITS1F (5-CTTGGTCATTTAGAGGAAGTAA-3′) and ITS2R (5-GTTTGCGTTCTTCATCGATGC-3′), buffer 4 µL, 2.5 mM dNTPs 2 µL, upstream primer (5uM) 0.8 µL, downstream primer (5uM) 0.8 µL, TransStart FastPfu DNA polymerase 0.4 µL, template DNA 10 ng and ddH2O to 20 µL. 3 replicates per sample (Wang et al., 2021).

Illumina Miseq sequencing

PCR products from the same sample were mixed and recovered on a 2% agarose gel. Recovered products were purified using the AxyPrep DNA Gel Extraction Kit (Axygen Biosciences, Union City, CA, USA), detected by electrophoresis on a 2% agarose gel, and analyzed with a Quantus™ Fluorometer (Promega). A fluorometer (Promega) was used to quantify the recovered product. Library construction was performed using the NEXTflexTM Rapid DNA-Seq Kit (PerkinElmer) by: (1) splice linkage; (2) removal of splice self-linked fragments using magnetic bead screening; (3) enrichment of library templates using PCR amplification; and (4) recovery of PCR products from magnetic beads to obtain the final library. Sequencing was performed using Illumina’s Miseq PE300 platform (Shanghai Meiji Biomedical Technology Co., Ltd., Shanghai, China) (Chen et al., 2016).

Statistical analysis

Shapiro–Wilk and Levene’s tests were used to verify the normality and homogeneity of the data. Data were mean ± standard deviation for three replicates. Two-way ANOVA and Tukey’s HSD post were used to analyze the differences among treatments. When the assumptions of normal distribution and homogeneity of variance are not satisfied, the non-parametric Kruskal-Wallis test is used to analyze the data, plotting by Origin and GraphPad prism. Fungal species annotation and assessment using mother (version v.1.30.2) index analysis, and species composition was based on the R language (version 3.3.1; R Core Team, 2016) vegan data package. Samples Qiime was used to calculate beta diversity distance matrices in comparative analyses, followed by tree drawing in R (version 3.3.1; R Core Team, 2016). Environmental factor association analyses were analyzed and mapped using RDA in the R language vegan package, and associated heat maps were used in R (heatmap package). Based on the PICRUSt2 function prediction package, predictions were made against the ITS amplicon sequencing results to obtain KO, pathway, and EC information based on information from the KEGG database and to calculate the abundance of each functional class based on OTU abundance. Additionally, for Pathway, PICRUSt was applied to obtain information on the three levels of metabolic pathways, and the abundance of each level was obtained separately.

Results

Analysis of root/soil environmental factors

Total root-soil chromium content increased with increasing Cr concentration throughout the treatment period, and total chromium content in the AMF addition group was higher than in the non-AMF addition group for the same Cr addition level. This difference gradually increased with increasing Cr concentration (6.81%, 9.4%, 10.81%, 16.54%, 28.97% from Group CK to Group D) (Fig. 1A). Soil Cr(III) content showed an increasing trend with increasing Cr(VI) concentration in both groups (Fig. 1B), indicating that Cr(VI) was constantly shifting to Cr(III) in the microenvironment. Root soil Cr(III) levels were increasing, and the shift from Cr(VI) to Cr(III) was facilitated by AMF addition. Root-soil SOM decreased with increasing Cr concentration in both groups, with the lowest SOM in the 200 mg/kg Cr(VI) group, and the highest SOM in the AMF addition group than in the non-AMF addition group at the same concentration (Fig. 1C). The control soil was weakly acidic and after adding potassium dichromate, remained so, and became more alkaline as Cr concentration increased (Fig. 1D). Under Cr (VI) stress, most plants without AMF addition were smaller and had yellow leaves, while plants with added AMF were larger, had more green leaves, and a more developed rhizosphere (Fig. S3).

Figure 1 The changes of environmental factors in the rhizosphere soils of Acorus calamus.

(A) Total Cr, (B) Cr (III) content, (C) SOM content and (D) pH inrhizosphere soils of Acorus calamus. Each group of data in the book is obtained by averaging three groups of parallel samples, and different lowercase letters of data in the same column indicate significant differences ( p < 0.05).

Species abundance of fungal community

To test whether differences between fungal groups were more significant than those within groups, ANOSIM analysis (Fig. 2A) was conducted using the Bray–Curtis distance algorithm. Between-group differences were considerably more significant than those within groups. Dilution curves after sampling are shown in Fig. 2B, and the amount of data sequenced was sufficient to reach the study level. PCA analysis used to investigate similarities or differences in the composition of the sample groups also showed that the coordinates clearly divided the AMF-added and non-AMF-added groups (Fig. 2D).

Figure 2 The comparative analysis of the samples.

(A) ANOSIM analysis. The X-axis is the distance value within or Between groups, the boxes corresponding to between represent the distance value between groups, and the other boxes represent the distance value within groups. The scale on the Y axis shows the magnitude of the distance value. It could test whether the difference between groups (two or more groups) is significantly greater than the difference within the group to determine whether the group is meaningful. (B) Dilution curve analysis. The abscissa represents the amount of randomly selected sequencing data. Ordinate, number of species observed. The amount of sequencing data is sufficient according to whether the curve is flat. (C) PCA analysis. The X-axis and Y-axis represent the two selected principal component axes, and the percentage represents the value of the principal component explaining the difference in sample composition. The red dots represent arbuscular mycorrhizal fungi (AMF) groups and the blue dots represent non-AMF groups. (D) PcoA analysis. The red dots represent AMF groups and the blue dots represent non-AMF groups.

The Venn diagram shows the number of OTUs common and unique to the added AMF and unadded AMF groups, as well as the similarity of OTU composition and overlap (Fig. 3). In the added AMF group, the number of overlapping OTUs was 132, and in each region the number also varied (Fig. 3A). In the group without the addition of AMF the number of overlapping OTUs was 133, with some variability in the number in each region compared to the group with the addition of AMF (Fig. 3B).

Figure 3 Venn graph.

The composition similarity and overlap of species (such as OTU) of environmental samples can be visually displayed. (A) AMF group and (B) non-AMF group.

According to the alpha analysis (Table S1) we found that the indices sobs, chao1 and ace, which characterize community richness, varied similarly. Fungal community richness was higher in all groups with AMF than in the control group, but fungal diversity was lower than in the non-AMF group. As Cr(VI) concentration increased, fungal community richness showed an overall decreasing trend, and there were significant differences among different concentrations. Moreover, fungal abundance in the Cr(VI) high concentration group was lower than the control group, and community abundance was lowest when Cr(VI) concentration reached 200 mg/kg. The same pattern can be seen in indices that characterize community diversity such as the Shannon and Simpson, where community diversity was lowest at Cr(VI) concentrations up to 200 mg/kg.

Fungal community species composition

Fungal species composition at phylum and genus levels differed significantly (Figs. 4A and 4B). The dominant phyla were Ascomycota, Basidiomycota, Mortierellomycota, Rozellomycota, Chytridiomycota, Blastocladiomycota, and Glomeromycota (Fig. 4A,). The percentage of dominant phyla in each group varied, but the common denominator was that the dominant species with the highest percentage were Ascomycota, accounting for 45.04%-87.43%. Talaomyces, Alternaria, Setophoma, Cladosporium, Fusarium and Botrytis represented Ascomycota at the genus level under Cr stress (Fig. 4B). The proportion of Ascomycota in the high concentration group (Cr =100–200 mg/kg) treatment was significantly higher than in the low concentration (Cr =50–100 mg/kg) treatment and control. In CK2 (with AMF added), Rozellomycota accounted for 27.94% of the dominant species, significantly higher than in the other groups. Blastocladiomycota was the most abundant in A2 (AMF added, Cr =10 mg/kg) at 17.06%. Fungal community abundance also differed between AMF treatments at the same Cr concentration, for example, the proportion of Ascomycota was significantly higher in the group with AMF addition than in the group without it. Mortierellomycota showed characteristics consistent with Ascomycota, decreasing in abundance with AMF addition. Basidiomycota, however, showed the opposite pattern to the two dominant species, as they increased significantly in the AMF addition group (except for the CK group) in groups A, B, C and D, by 70.31%, 83.99%, 71.51%, and 84.55%, respectively.

Figure 4 The community plot of the fungal species.

(A) Bar diagram of fungal community under phylum classification, (B) bar diagram of fungal community under genus classification.

To further explore the proportion of different dominant species in each group of samples, as well as the distribution ratio of each dominant species in different subgroups, two circle diagrams (Fig. 5), were constructed to represent fungal species distributions at phylum and genus levels. Ascomycota were evenly distributed across treatments, exceeding 70%, and did not significantly change with increasing Cr concentration. However, the proportion of Ascomycota in the AMF-added group was lower than at the same Cr concentration in the non-AMF-added group. Mortierellomycota had the highest proportion of unspiked A1, followed by a sharp drop in Cr concentration up to 50 mg/kg and then a slight rebound at Cr = 200 mg/kg but not back to the unspiked CK1 vs. A1 proportion. Basidiomycota was highest in the A1 group (CCr = 10 mg/kg) and then showed a decreasing trend with increasing Cr concentration. Chytridiomycota distribution in the different groups fluctuated with increasing Cr concentration, and their percentage decreased sharply when Cr concentration increased from 50 mg/kg to 100 mg/kg and then increased sharply when it reached 200 mg/kg. Glomeromycota percentage was highest in the low concentration groups (CK1, A1) and lower in the high concentration groups (B1, C1, D1).

Figure 5 Circos diagram of fungal community.

Figure reflected the distribution proportion of dominant species in each sample and the distribution proportion of each dominant species in different samples. (A) Circos diagram of fungal community under phylum classification and (B) genus classification.

With AMF addition, the proportion of the dominant clade changed significantly. For example, Mortierellomycota decreased from the second to the 5th position, while Blastocladiomycota increased from last to 4th. Ascomycota proportion in each sample group and in different subgroups remained the highest, and increased with increasing Cr concentration until it reached a peak at 100 mg/kg, and then decreased when Cr concentration reached 200 mg/kg. Rozellomycota remained the most abundant in the CK2 group, with a significant increase when Cr concentration was 200 mg/kg compared to the non-AMF group, and a significant decrease when Cr concentration was 50 mg/kg. Blastocladiomycota distribution characteristics did not change significantly, however, the distribution ratio in different groups increased significantly. The distribution ratio of Mortierellomycota reached a maximum in group CK2, and a minimum in B2 (Cr concentration of 50 mg/kg) and D2 (Cr concentration of 200 mg/kg). After AMF addition, the highest Chytridiomycota percentage was still found in the D2 group when Cr concentration was 200 mg/kg. Fungal composition also changed significantly at the genus level (Fig. 5B).

Environmental factor association analysis

To assess the correlation between the soil microenvironment and the structural classification of the fungal community at the phylum and genus level in the AMF addition and control groups, correlation heatmap plots were made based on the spearman analysis index, and these show the top 40 species in terms of total abundance at the genus level (Figs. 6 and 7). Correlations between fungi and pH, SOM, Cr, Cr(VI) at the phylum level changed after AMF addition (Fig. 6). The dominant fungi at the genus level and their correlations with the above factors also changed significantly (Fig. 7A). Of these, Cr(VI) and total Cr correlations with fungal species were significantly altered. Before AMF addition, Cr was negative correlated with Atractium (Ascomycota) but positively with Arthrobotrys (Ascomycota) with a correlation coefficient of 0.637 (p < 0.05). Similarly, Cr(VI) was negatively correlated with Saitozyma (Basidiomycota) (p < 0.01) but positively correlated with Setophoma (Ascomycota) (p < 0.01), presumably because Ascomycota may contain fungi with a corresponding resistance mechanism to Cr stress. After the addition of AMF (Fig. 7B), it was clear that the number of fungi positively correlated with Cr, Cr(VI) increased significantly, with unclassified Pleosporales significantly correlated (p < 0.01). Cr was positively correlated with Botrytis (Ascomycota) and Ascobolus (Ascomycota), and before the addition of AMF showed significant negative correlations with Ascobolus, and Cr(VI) showed positive correlations with Phlyctochytrium (Chytridiomycota), Didymella (Ascomycota), Botrytis, Setophoma, and Acremonium (Ascomycota) showed significant positive correlations, and the change in the correlations revealed that the correlation between some fungi and total chromium changed from negative to positive. Meanwhile there was a significant increase in the number of fungi positively correlated with Cr(VI). Overall it appears that AMF addition: (1) improved Cr resistance of some fungi (mainly Ascomycota) to a certain extent, and (2) promoted the ability of fungi to reduce Cr(VI) to Cr(III).

Figure 6 The association analysis diagram of environmental factors.

(A) The correlation plots of the measured factors (SOM, pH, Cr, Cr (VI)) and the fungal communities at the phylum level in AMF group; (B) the correlation plots of the measured factors and the fungal communities at the phylum level in the non-AMF group.

Figure 7 The association analysis diagram of environmental factors.

(A) The correlation plots of the measured factors and the fungal communities at the genus level in the AMF group; (B) the correlation plots of the measured factors and the fungal communities at the genus level in the non-AMF group.

Functional analysis of fungal community

Electrons are transferred from NADH to ubiquinone by the action of dehydrogenase, after which they flow to cytochrome c, and then Cr(VI) receives electrons from cytochrome c to be reduced to Cr(III) (Xia et al., 2018). Fungal abundance values in the cytochrome c pathway showed an overall increase with increasing Cr(VI) concentrations (Fig. 8A), demonstrating that the cytochrome c pathway species increased under Cr stress and became more pronounced with AMF addition. In the AMF group, NADH dehydrogenase fungal abundance increased overall. Species abundance in the NADH dehydrogenase pathway was higher in the AMF group than in the control group (Fig. 8B), and showed an increasing trend in ubiquinone reductase (H(+)-translocating), also more significant in the AMF group than in the control group (Fig. 8C). As an organism essential cycle of energy and material metabolism, TCA uses organic carbon sources (mainly glucose) to produce energy through oxidation by the gluconeogenic pathway. Under Cr(VI) stress, TCA pathway fungal abundance was significantly higher after AMF addition than in the non-AMF group (Fig. 8D). It is hypothesized that the remaining fungal tricarboxylic acid cycle was facilitated by the effect of AMF, with increased use of carbon sources and enhanced energy metabolism.

Figure 8 The functional analysis map of fungal community.

(A) Cytochrome c pathway; (B) NADH dehydrogenase; (C) NADH: ubiquiinone reductase (H + -translocating); (D) TCA cycle. The Y-axis shows the fungal abundance value of each group.

Discussion

Under heavy metal stress, microorganisms are more sensitive than other organisms (Charlton et al., 2016; Giller, Witter & McGrath, 2009), and their abundance is altered in various ways. For example, Flavisolbacter and Altererythrobacter abundance is affected by Cr stress (Zhang et al., 2020). Furthermore, fungal community composition was significantly altered in response to changes in heavy metal content (Del Val, Barea & Azcón-Aguilar, 1999; Nordgren et al., 1983). In our study, fungal diversity and abundance were significantly altered by Cr(VI) stress, and overall they showed a decreasing trend with increasing Cr(VI) concentrations in both control and AMF groups. In long-term chronically Cr-contaminated soils, fungal abundance declines but fungal community diversity is not altered (Jin et al., 2018). We showed significant changes in fungal community abundance and diversity at various concentrations of Cr(VI) stress. However, fungal community abundance showed an overall decreasing trend with increasing Cr(VI), especially at high concentrations.

Fungal community species composition changed significantly under Cr stress, and dominant group distribution ratios also changed. The most dominant species under Cr(VI) stress were Ascomycota, followed by Basidiomycota fungi, and high Cr concentration increased Ascomycota fungi. In the correlation heat map, Ascomycota fungi were significantly correlated with Cr(VI) and Cr, so it may be that these fungi, along with some in the Basidiomycota, are sensitive, or resistant to chromium. Derxomyces (Basidiomycota) is very sensitive to Cr and shows a significant negative correlation with Cr and Arthrinium (Jin et al., 2018). Moreover, some Ascomycota fungi can reduce Cr(VI) to Cr(III) using carbon metabolism capabilities, such as Aspergillus sp., Penicillium sp., and Trichoderma hamatum (Acevedo-Aguilar et al., 2006). Lazarova et al. (2014) found that Trichosporon (Ascomycota) was able to grow under 10 mmol/LCr stress and was highly resistant to chromium because it can chelate and even reduce Cr(VI) from heavy metal ions (Georgieva et al., 2011; Bajgai, Georgieva & Lazarova, 2012). Candida is also highly resistant to Cr and can survive in media containing 100 mmol/L Cr stress due to the presence of chromate reductase (Ramírez-Ramírez et al., 2004). Aspergillus (Ascomycota) is also a common chromium-resistant fungus that reduces Cr(VI) to Cr(III) (CzakóVér et al., 1999), and responds to Cr contamination mainly by enriching in vivo (Mala, Nair & Puvanakrishnan, 2006). Coreno-Alonso et al. (2019) found that the Ed8 strain of Aspergillus tubingensis reduced Cr(VI) concentration through reduction reactions stimulated by carboxylic acids and metal chelators.

Fungal reduction of Cr(VI) reduction can be in vivo as well as in vitro(Fig. 9). The conversion of hexavalent chromium to trivalent chromium is through an electron reduction reaction. Microorganisms generally convert hexavalent chromium to trivalent chromium through enzyme and non-enzyme reductions that perform an electron transfer role, transferring 3 electrons to hexavalent chromium. Chromate first enters the fungus through the sulfate pathway due to its similar chemical structure to sulfate, and then a Cr(VI) portion is reduced to Cr(III) through enzymatic and non-enzymatic reduction (non-enzymatic reducing substances such as GSH and cysteine) (Thatoi et al., 2014). Soluble reductases are dominated by ChrR, Yief, and NfoR, and their enzymatic reductions occur under aerobic conditions (Eswaramoorthy et al., 2012; Ackerley et al., 2004; He et al., 2011; Han et al., 2017). The CHR-1 protein, homologous to chrA, is found in a number of Cysticercus, Streptomycetes, and Seamycetes fungi. CHR-1 not only reduces Cr(VI) and immobilizes Cr in fungal vesicles, but chrA is found in many microorganisms as an efflux protein and is an additional measure of resistance to Cr(VI) (Viti et al., 2014).

Figure 9 The proposed mechanism of Cr(VI) resistance with the addition of AMF increases the root-soil SOC content and thus fungal activity.

(I) Chromate enters through the sulfate channel; (II) Cr(VI) is reduced to Cr(III) through enzymatic and non-enzymatic reduction and subsequently immobilized in the vesicle via CHR-1; (III) Cr(VI) is reduced in vitro by the fungus through electron transfer; (IV) Chromate is exocytosed through chr-A; (V) The mucus adsorption on the fungal surface, as well as increased protonation levels at low pH, attract chromate ions, which are reduced and then released through electron repulsion release.

In addition to the mucus produced by fungi in vitro to adsorb heavy metal ions, fungal Cr(VI) reduction in vitro is mainly associated with electron transfer. Low pH is also beneficial for Cr(VI) reduction (Martorell et al., 2012). In our study, the soil was weakly acidic, which increases the fungus surface protonation level, making the positively charged surface better able to attract negatively charged chromate ions, which are thereafter reduced to Cr(III) due to electron repulsion release (Park et al., 2005). Xia et al. (2018)) found that Cr(VI) ground electron reduction was achieved by transferring them from NADH to ubiquinone in the presence of dehydrogenase CymA, MtrA, MtrB, MtrC, and OmcA, which are all cytochrome c. We found that Cr(VI) addition added to the fungal cytochrome c metabolic pathway, NADH dehydrogenase, and NADH: ubiquinone reductase, demonstrating that the fungus has in vitro reductive effects under Cr(VI) stress.

Fungi are heterotrophic organisms that use soil organic matter as their primary carbon source (Lehmann & Kleber, 2015; Xue et al., 2018), however, their ability to utilize carbon sources is limited at high heavy metal concentrations (Georgieva et al., 2011). However, energy metabolism plays a vital role in the fungal response against heavy metals (Jin et al., 2008). Gasch & Wemer-Washburne (2002) found that many genes involved in carbohydrate and fatty acid metabolism in Common Metal Responsive (CMR) were upregulated in response to heavy metal stress. Moreover, Acevedo-Aguilar et al. (2006) showed that Cr(VI) reduction to Cr(III) by fungi requires a carbon source that is fermented like glucose or oxidized like glycerol, and that total Cr(VI) is unaltered in the absence of a carbon source. AMF can have a beneficial effect on lowering plant exposure to heavy metals by improving water and soil nutrient uptake, increasing aboveground biomass and causing changes in root morphology, reducing oxidative stress from heavy metals (Muhammad et al., 2020). Furthermore, increasing soil SOM with AMF forms fibrous roots with plants, a change that may somewhat improve Cr tolerance of fungal community. The main reason for the increase in soil SOM is that AMF can secrete glycoproteins, such as the globulin-related soil protein (GRSP), which plays a significant role in rhizosphere soil aggregation, carbon storage, and soil quality improvement (Nichols, 2003; Rillig et al., 2001; Schüßler, Schwarzot & Walker, 2001) and can protect AMF mycelia from nutrient loss (Wessels, 1996). AMF and GRSP play crucial roles in direct and indirect soil carbon sequestration. Directly, AMF increases soil carbon content through increased plant growth and aboveground biomass, in the context of root and root deposition input. Indirectly, GRSP improves carbon sequestration (Miller, Reinhardt & Jastrow, 1995) through soil aggregation, forming stable agglomerates that protect organic compounds as well as microbial necromass from enzymatic and microbial attack (Awad et al., 2013). Furthermore, GRSP can increase the active carbon pool by increasing microbial activity (Subramanian et al., 2019; Wright & Upadhyaya, 1998). In our study, AMF addition increased SOM content (Fig. 1C), improved the root system soil microenvironment, improved the limitation of fungal access to carbon sources, increased the fungi metabolic level in the AMF group (Fig. 8D), increased fungal ability to access carbon sources, and increased their own ATP synthesis, which was also beneficial for Cr(VI) reduction.

Conclusion

Fungal community abundance and diversity were suppressed under Cr stress, especially at high Cr(VI) concentrations (Cr(VI) = 100 mg/kg-200 mg/kg). The dominant species were Ascomycota, Basidiomycota, and Mortierellomycota. Fungi with high tolerance to Cr contamination were present in the Cysticercus phylum. After AMF addition, SOM content increased, conversion of Cr(VI) to Cr(III) increased, fungal abundance increased, and fungal diversity increased in the higher concentration groups. The fungal community has a corresponding resistance mechanism under Cr(VI) stress, as Cr(VI) can reduce the metabolic capacity of the fungal community. Cr(VI) reduces root-soil SOM, thus affecting fungal community change, however, AMF increases root-soil SOM content, which also improves the Cr(VI) resistance capacity of fungal community and its ability to enhance Cr(VI) reduction.

Overall, there were significant differences in fungal community changes under different concentrations of Cr (VI), and fungal abundance increased after AMF addition. In contrast, fungal diversity increased at high Cr (VI) concentrations. AMF addition enhanced fungal community Cr(VI) reduction, but the specific fungus responsible and the principal reduction mechanism remain undetermined.

Supplemental Information

Supplemental Information 1 AMF increases root-soil SOM content, which also improves the Cr(VI) resistance capacity of the fungal community and its ability to enhance Cr(VI) reduction and promote plant growth under Cr (VI) stress

Click here for additional data file.

Supplemental Information 2 Table S1 Alpha diversity analysis table

Alpha diversity analysis reflects the richness and diversity of microbial communities, including a series of statistical analysis indexes to estimate the species abundance and diversity of ecological communities.The indexes reflecting the Community richness include: Sobs, Chao1, Ace.The indexes reflecting the Community diversity include: Shannon, Simpson.

Click here for additional data file.

Supplemental Information 3 Acorus calamus pot experiment design

Experimental groups were divided into AMF-added, and non-AMF-added, each with five Cr levels: 0, 10, 50, 100, 200 mg/kg respectively corresponded to CK, A, B, C, D group, and three parallel groups for each treatment.

Click here for additional data file.

Supplemental Information 4 AMF colonization map

Successful colonization of arbuscular mycorrhizal fungi under electron microscope

Click here for additional data file.

Supplemental Information 5 Plant growth map

Comparison of growth of Acorus calamus under Cr stress. Left:the plants without the addition of AMF; Right:the plants with the addition of AMF

Click here for additional data file.

Supplemental Information 6 The raw data for soil properties

Click here for additional data file.

The authors would like to express their gratitude to EditSprings for the expert linguistic services provided.

Additional Information and Declarations

Competing Interests

Author Contributions

Data Availability

The authors declare there are no competing interests.

Guodong Xia conceived and designed the experiments, performed the experiments, analyzed the data, prepared figures and/or tables, authored or reviewed drafts of the article, and approved the final draft.

Sixi Zhu conceived and designed the experiments, authored or reviewed drafts of the article, and approved the final draft.

Wei Zhao analyzed the data, authored or reviewed drafts of the article, and approved the final draft.

Xiuqing Yang analyzed the data, authored or reviewed drafts of the article, and approved the final draft.

Luying Sheng analyzed the data, prepared figures and/or tables, and approved the final draft.

Huan Mao analyzed the data, prepared figures and/or tables, and approved the final draft.

The following information was supplied regarding data availability:

The sequence reads are available at BioProject: PRJNA952550.

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
