# Peer review of "Arbuscular mycorrhizal fungi alter rhizosphere fungal community characteristics of Acorus calamus to improve Cr resistance"

_PeerJ, doi:10.7717/peerj.15681_

## Round 0.1 · original submission · Minor Revisions

Two experts assessed your manuscript and found it interesting enough to be published in this journal, after addressing the raised concerns. These include improvement of readability, and the addition of references to support the methodologies.

Reviewer 1 ·

Basic reporting

The manuscript is well structured. However, some methods described do not have the appropriate references.

Raw data has been shown in the supplementary material section.

Experimental design

The focus of the study, as well as the methods used to discover the ability of fungal communities to reduce Cr VI from contaminated sites, is adequate. The methods are well described, but as I commented above, they are required to be referenced.

Validity of the findings

In some sections of the discussion of the manuscript, it seems that it is contradicted, since it is mentioned that in the study there were no significant differences regarding the abundance and diversity of the fungal community at various concentrations of Cr.
But in the next paragraph, he mentions the dominance of one species over the others under the stress of Cr.

I suggest you rearrange the ideas so that potential readers are not confused and so that this part of the manuscript has coherence.

(lines: 302-315)


The first paragraph of the conclusions should be improved since the ideas shown here contradict each other.

I suggest you rewrite it to clarify what is intended to be understood considering the results shown.

Additional comments

Line 10: Define SOM.

Lines 55- 57: the chemical writing of the chromates must be corrected (use superscripts)

Line 64: Define CWs


In some parts of the manuscript, the microorganisms' names are not in italics.

Line 109: Reference of the method.

Lines 116-124: ibidem

Line 127: Define EC. (In general, throughout the text there are several acronyms or initials that must be defined).


Lines 344-348: How to demonstrate the participation of these enzymes in the reduction of Cr VI; Is there gene overexpression analysis?


I suggest you add as supplementary material, images of the plants where you can see the difference between the addition of AMF or its absence in the different concentrations of Cr VI.

Reviewer 2 ·

Basic reporting

Chromium (Cr) can contaminate surface water, groundwater, etc. With the rapid development of industry, Cr pollutants accumulate constantly in the ground, causing severe soil contamination problems. Farmland Cr pollution hurts the safety of agricultural production, and it directly or indirectly affects human health and safety. To render the contaminated resource reusable, Cr must be removed physically or by using some reliable techniques.
In the article, “Arbuscular mycorrhizal fungi alter rhizosphere fungal community characteristics of Acorus calamus to improve Cr resistance”, the author aimed to compare fungal community structure and functional properties in an arbuscular mycorrhizal fungi (AMF) group with a control group. A high-throughput sequencing technique was used to characterize the fungal community’s response under Cr stress conditions.
In my opinion, the article is well-written and educational. However, I have minor concerns given below.

Minor concerns:

There are errors, spelling mistakes in the manuscript as well as in figures that should be corrected before publication.
Line 117: Include the word concentration between Cr and levels.
Line 124: Spelling mistake in Hoagland’s solution.
Line 139 and 140: Include the proper symbol for micromolar.
Line 174-176: It is not clear; it should match with the given figure.
Line 193, 195 and 255; write the word figure in the same format (either Figure or Fig.)
Remove the word were from the Figure 4 legend.
Keep the correlation plots of the measured factors (SOM, pH, Cr, Cr (VI)) in Figure 7b in the same order as Figure 7a.
Spelling mistakes in figure legend: Figures 6 and 7; non-AMF
Spelling mistakes: figure 9; Ubiquinone, Cr(VI) reductase and CHR-1. Also, can you explain how it became K2CrO4 from K2Cr2O7?

Experimental design

No any improvement required.

Validity of the findings

no comment.

Additional comments

no comment.

---

## Round 0.2 · accepted · Accept

The authors properly addressed the Reviewers' concerns; consequently, their manuscript is suitable for publication.